# Multi-scale Stochastic Generation of Labelled Microscopy Images for Neuron Segmentation

**Meghane Decroocq**[1]                                    MEGHANE.DECROOCQ@RIKEN.JP
[1] *Brain Image Analysis Unit, Center for Brain Science, RIKEN, Wako, Japan*

**Binbin Xu**[2]                                          BINBIN.XU@MINES-ALES.FR
[2] *EuroMov Digital Health in Motion, Univ Montpellier, IMT Mines Ales, France*

**Katherine L Thompson-Peer**[3]                           KTPEER@UCI.EDU
[3] *Department of Developmental and Cell Biology, University of California, Irvine, CA 92697, USA*

**Adrian Moore**[4]                                        ADRIAN.MOORE@RIKEN.JP
[4] *Laboratory for Neurodiversity, Center for Brain Science, RIKEN Wako, Japan*

**Henrik Skibbe**[*1]                                      HENRIK.SKIBBE@RIKEN.JP

**Editors:** Accepted for publication at MIDL 2024

## Abstract

We introduce a novel method leveraging conditional generative adversarial networks (cGANs) to generate diverse, high-resolution microscopy images for neuron tracing model training. This approach addresses the challenge of limited annotated data availability, a significant obstacle in automating neuron dendrite tracing. Our technique utilizes a multi-scale cascade process to generate synthetic images from single neuron tractograms, accurately replicating the complex characteristics of real microscopy images, encompassing imaging artifacts and background structures. In experiments, our method generates diverse images that mimic the characteristics of two distinct neuron microscopy datasets, which were successfully used as training data in the segmentation task of real neuron images.

**Keywords:** multi-scale cGAN, microscopy imaging, neuron tracing

## 1. Introduction

The automated segmentation and tracing of neuronal structures from microscopy images is a crucial step toward understanding brain structure and function. Despite its importance, it remains a challenging task that requires high levels of morphological precision, often necessitating time-consuming manual efforts. Recent advancements in deep learning have led to substantial improvements in automatic neuron segmentation and tracing (Chen et al., 2021). However, the effectiveness of these models is constrained by the scarcity of annotated training data and the diversity of imaging methods and neuronal morphologies.

One approach to circumvent the limitations associated with the scarcity of annotated data is the generation of artificial images. This involves rasterizing a tree structure into an image and adding noise to simulate real-world imperfections. Such synthetic images are often utilized to expand existing training datasets or even to directly train models, thereby rendering the time-consuming acquisition and labeling of real images obsolete (Chen et al., 2021). This technique is not limited to neuron segmentation but also benefits the analysis of other tree-like structures, such as vessels (Tetteh et al., 2020; Prabhakar et al., 2024) and

airways (Nardelli et al., 2020). To create images that mimic realistic morphology, these are often generated from publicly available digital reconstructions (e.g., neuron tractograms, vascular trees) (Radojević and Meijering, 2019; Chen et al., 2021), or artificially generated trees (Hamarneh and Jassi, 2010). Synthetic images can be generated in substantial volumes, effectively addressing the scarcity of real-world data and the imprecision of manual annotations. However, they often fail to accurately replicate the surrounding anatomical structures or imaging artifacts unique to specific organs and imaging modalities, thus limiting the transferability of the model to real-world images.

In recent years, generative adversarial networks (GAN), particularly conditional GANs (cGANs) (Isola et al., 2017), have demonstrated remarkable capabilities in generating realistic synthetic images. These models generate images conditioned on provided ground truth, creating paired images that bypass the need for further annotation. cGANs have been successfully employed across several imaging modalities, including CT (Jin et al., 2018), MR (Lau et al., 2018; Mok and Chung, 2019), ultrasound (Tom and Sheet, 2018), and retinal images (Costa et al., 2017). However, their application to microscopy imaging of neurons, to the best of our knowledge, remains unexplored. The unique challenges posed by neuron imaging include the high resolution of microscopy images, which imposes substantial computational demand and necessitates a large amount of annotated data for training. Moreover, the one-to-one image mapping inherited from cGANs constrains the diversity of the generated images to that of the input ground truth images. Consequently, achieving a broad range of image outputs requires a large and varied dataset of neuron tractograms.

To address these challenges, we propose a novel method utilizing cGANs. Our approach leverages a multi-scale cGAN cascade to iteratively refine image resolution, employing a patch-based strategy that facilitates the generation of high-resolution microscopy images with minimal memory demands. An important part of our approach is the employment of a mode seeking loss function (Mao et al., 2019) that enhances the diversity of the generated data, setting our method apart from previous methods (Uzunova et al., 2019) that generate unique pairs of real and generated images. We have developed a new training strategy that ensures coherence of the image content across different scales during refinement. This allows for the stochastic generation of a broader variety of images without compromising quality. We have successfully applied our method to two datasets of neuron microscopy images, producing synthetic images that faithfully replicate the characteristics of their real counterparts. The code is available at https://github.com/BrainImageAnalysis/Multi-scale-cGAN.

## 2. Methods

### 2.1. Data

In this study, we use two datasets of *Drosophila* single-neuron microscopy images with diverse anatomical structures, and real-world imaging artifacts. Dataset 1 contains 417 widefield fluorescent microscopy images of *Drosophila* class I da neurons. Dataset 2 (Thompson-Peer et al., 2016; Nguyen and Thompson-Peer, 2021) contains 221 confocal microscopy images of *Drosophila* class IV da neurons; see Appendix A for further details. The Z-stack images were converted to 2D by maximum intensity projection, and the images of both datasets have been rescaled from $1920 \times 1440$ and $1024 \times 1024$ pixels to $1024 \times 1024$ pixels.

Both datasets were manually traced. Neurite radii and the cell bodies were automatically extracted. All images were post-processed using histogram equalization. We downloaded all the available tracings for *Drosophila* class I and IV da neurons from the NeuroMorpho database (Ascoli et al., 2007), and rasterized them into a $1024 \times 1024$ segmentation mask, forming the evaluation datasets Neuromorpho 1 (373 segmentations) and Neuromorpho 2 (501 segmentations).

## 2.2. Image generation

### 2.2.1. MULTI-SCALE CASCADE STRUCTURE

Models such as Laplacian GAN (Denton et al., 2015) or HDpix2pix (Wang et al., 2018) employ a multi-scale iterative refinement to generate high-resolution images. However, these methods impose high memory demands by requiring entire images as inputs to the network. To circumvent this limitation, we utilize a cascade of conditional GANs with a fixed input and output size. As proposed in (Uzunova et al., 2019), rather than enlarging the size of the image processed through the model, we keep the patch size constant while iteratively decreasing the receptive field, thereby increasing the pixel resolution (Figure 1).

Conditional GANs are generative models that learn to map a condition image $x$ to a target image $y$, $G : x \to y$. In this work, $x$ is the neuron segmentation and $y$ is the paired microscopy image. A discriminator $D$ is simultaneously trained to distinguish fake from real images, driving the generator to create realistic images while following the constraints of the input image $x$.

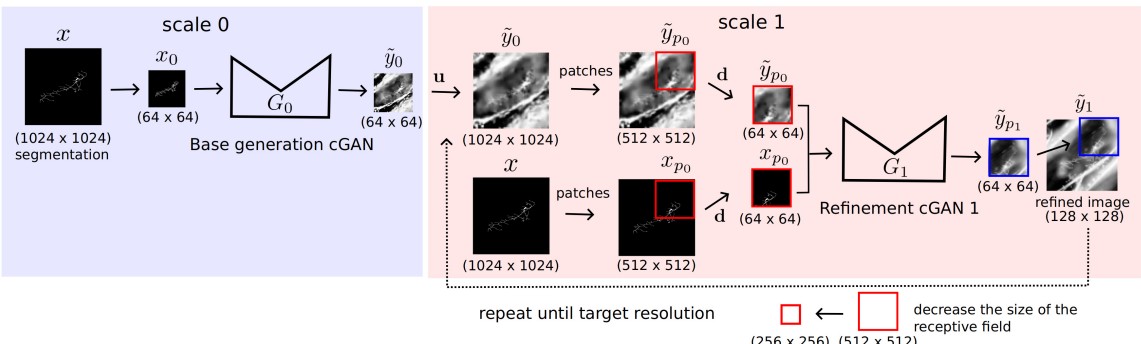

**Figure 1:** Multi-scale generation pipeline. **u** (respectively **d**) refers to an upsampling (respectively downsampling) operation, and $\tilde{y}_{p_0}$ (respectively $x_{p_0}$) designates the patches extracted from the whole image $\tilde{y}_0$ (respectively $x_0$).

At the first scale, a conditional GAN $(G_0, D_0)$ learns the mapping between a low-resolution $(64 \times 64)$ neuron segmentation image $x_0$ and a low-resolution microscopy image $y_0$, $G_0 : x_0 \to \tilde{y}_0$. The generated image $G_0(x_0)$ is upsampled to the original resolution $(1024 \times 1024)$ and divided into patches $\tilde{y}_{p_0}$. In the next step, another conditional GAN $(G_1, D_1)$ learns to generate next resolution patch $\tilde{y}_{p_1}$ conditioned on both the segmentation $x_{p_0}$ and the previous patch $\tilde{y}_{p_0}$, $G_1 : (x_{p_1}, \tilde{y}_{p_0}) \to \tilde{y}_{p_1}$. The generated patches are combined and upsampled to the original resolution to form the refined images $\tilde{y}_1$. This refinement step is repeated until we reach the target resolution. At every scale, the receptive field of the patches is divided by two; the resolution of the refined images is consequently doubled.

The objective of the base generative cGAN $(G_0, D_0)$ and the refinement cGANs $(G_i, D_i)$ with $i \in [1, n]$ differs. We detail the architecture of each model in the next sections.

### 2.2.2. Base generation cGAN

The objective of the base generative model is to create a coarse image background, laying the foundational layer for subsequent refinement at higher scales. This base model is crucial, as it determines both the realism and the diversity of the generated images.

cGANs usually suffer from the lack of diversity of the generated samples for a given input, primarily because the model tends to overlook the random noise vectors. This oversight leads to a predominant one-to-one mapping between the input condition and the target image. The widely used cGAN-based pix2pix model (Isola et al., 2017) relies on a $L_1$ loss between the generated image $G_0(x_0)$ and the target image $y_0$ to improve the realism of the generated images, restricting even more the variability of the generated samples. In this type of model, the only source of variability is the input image $x$. To increase the diversity of the generated images, Mao et al. (Mao et al., 2019) proposed a new regularization term to drive the generator to generate dissimilar images, called mode seeking loss. We propose to use this loss in our base generation model to increase the diversity of the generated images.

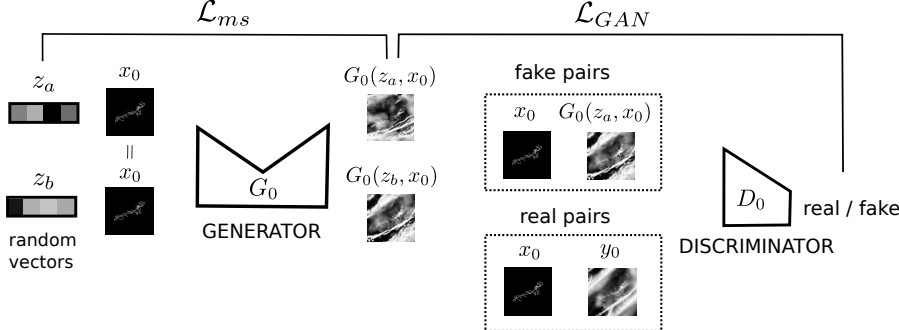

**Figure 2:** Mode seeking conditional GAN model.

Our base generation cGAN is composed of a generator $G_0$ and a discriminator $D_0$. The generator is a U-Net (Ronneberger et al., 2015), and the discriminator a PatchGAN (Isola et al., 2017). The generator $G_0$ takes as input the downsampled neuron segmentation image $x_0$ and a random vector $z$ drawn from a uniform distribution. The discriminator $D_0$ is trained to distinguish fake pairs $(x_0, G(z, x0))$ from real pairs $(x_0, y_0)$. As shown in Figure 2, we use two losses during the training. The mode seeking loss $\mathcal{L}_{ms}$ (Equation 1) maximizes the dissimilarity between two images generated from different $z$ random vectors, increasing the diversity of the generated images and avoiding mode collapse. The adversarial loss $\mathcal{L}_{GAN}$ (Equation 2) pushes the generator to create realistic images. These losses are:

$$\mathcal{L}_{ms} = \max_{G_0}\left(\frac{MAE(G_0(z_a,x_0),G_0(z_b,x_0))}{MAE(z_a,z_b)}\right), \tag{1}$$

where $z_a$ and $z_b$ are two random vectors of size $n_z$, and $MAE$ is the mean absolute error, and

$$\mathcal{L}_{GAN}(G_0, D_0) = \min_{G_0}\max_{D_0}\mathbb{E}_{x_0,y_0}[\log D_0(x_0, y_0)] + \mathbb{E}_{z,x_0}[1 - \log D_0(x_0, G_0(z, x_0))]. \tag{2}$$

The total loss $\mathcal{L}$ is the sum of both losses $\mathcal{L} = \mathcal{L}_{ms} + \mathcal{L}_{GAN}$. Figure 3 shows the effect of the mode seeking loss on the diversity of the generated images compared to the original pix2pix model.

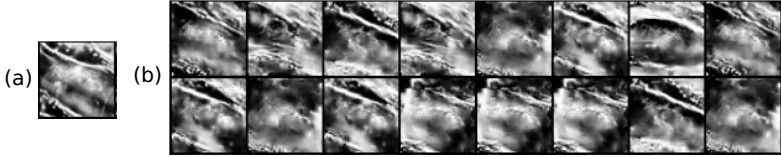

**Figure 3:** Repeated generation for the same input tractogram, using pix2pix (a), and mode seeking (b). As pix2pix repeatedly generated the same image, we show only one of them.

### 2.2.3. REFINEMENT CGAN

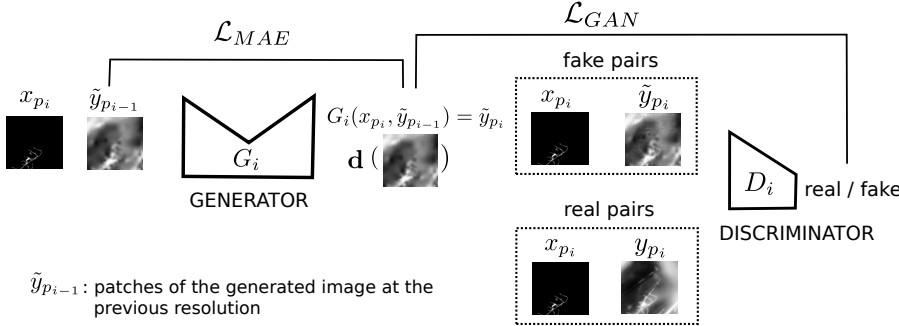

**Figure 4:** Refinement conditional GAN model.

The low-resolution images produced by the base generation cGAN are progressively enhanced in quality through a cascade of refinement cGAN models at subsequent scales. The structure of our refinement cGAN is illustrated in Figure 4. We keep the same architecture as the base generation cGAN for the generator $G_i$ and the discriminator $D_i$.

The stochasticity of our base generation cGAN drove us to develop a new training strategy for refinement compared to existing methods. Unlike the multi-scale GAN of (Uzunova et al., 2019), the images produced by the base cGAN do not correspond to real images. Therefore, they cannot be used as matched pairs for training the discriminator. To prevent the generator from ignoring the image generated at the previous scale (a role normally undertaken by the discriminator in cGANs), we use an $MAE$ loss between the generated images at scales $i - 1$ and $i$;

$$\mathcal{L}_{MAE}(\tilde{y}_{p_{i-1}}, \tilde{y}_{p_i}) = MAE(\tilde{y}_{p_{i-1}}, \mathbf{d}(\tilde{y}_{p_i})), \tag{3}$$

where $\mathbf{d}$ is the operation of downsampling to the resolution of the previous scale, and $\tilde{y}_{p_i}$ are the patches generated at scale $i$. The adversarial loss can be written as follows:

$$\mathcal{L}_{GAN}(G_i, D_i) = \min_{G_i} \max_{D_i} 2\mathbb{E}_{x_{p_i}, y_{p_i}}[\log D_i(x_{p_i}, y_{p_i})] + \mathbb{E}_{x_{p_i}, y_{p_{i-1}}}[1 - \log D_i(x_{p_i}, \tilde{y}_{p_i})] \tag{4}$$

The total loss $\mathcal{L}$ is a combination of the adversarial loss and the $L_1$ losses, balanced by a parameter $\lambda$; $\mathcal{L} = \mathcal{L}_{GAN}(G_i, D_i) + \lambda\mathcal{L}_{MAE}(\tilde{y}_{p_{i-1}}, \tilde{y}_{p_i})$.

### 2.2.4. TRAINING DETAILS

Data augmentation (90 degree clockwise / counterclockwise rotations, horizontal and vertical flip) is applied to the images only during the base generation of scale 0. The model is trained with a batch size of 32, a learning rate of 0.002 using an Adam optimizer for all scales. Following heuristic exploration, the length of the random $z$ vectors is set to $n_z = 12$, and the $\lambda$ parameter to 10. During the application of our method to the microscopy Datasets 1 and 2, we observed that keeping a model input patch size of $64 \times 64$ in the high resolutions of the cascade led to the creation of artifacts in the background of the image. We believe that it is caused by the lack of context of the patches in the high resolutions. Increasing the size of the patches to input in the model in the high-resolutions solved this problem while keeping the memory demand reasonable. The best trade-off parameters, as well as the number of iterations for each scale, can be found in Table 3. For training and inference, we divide the images into regularly organized patches with a 50% overlap. The overlapping parts of the generated patches are merged using cosine weights, to avoid border effects in the generated images.

## 3. Results

### 3.1. Image generation

We present visual examples of images generated with our model. Figure 5 illustrates the ability of the refinement cGAN to enhance the details of the rather coarse image generated at scale 0. As shown in Figure 6, our method can reproduce characteristics of the real microscopy images, such as background structures, including other surrounding neurons in Dataset 2, and imaging artifacts. Moreover, as illustrated in Figure 5, our generative model can produce a diverse set of images from a single input neuron tractogram, while maintaining the morphology of the input neuron (more examples in Appendix D).

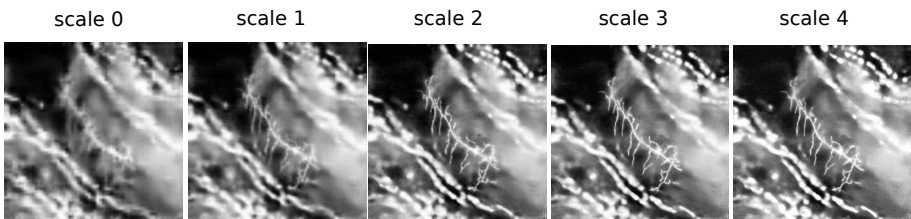

**Figure 5:** Images generated at the different scales, from coarse to fine resolution.

### 3.2. Ablation study

In this section, we emphasize the contribution of the different components of our model through an ablation study. To this aim, we evaluated five models of increasing complexity; (1) pix2pix model (Isola et al., 2017), refined according to the method of (Uzunova et al., 2019), where the images generated at the previous scale are shown to the discriminator, (2) a hybrid model where the cGAN with mode seeking loss proposed in this work is used at

generated images        segmentation  original image

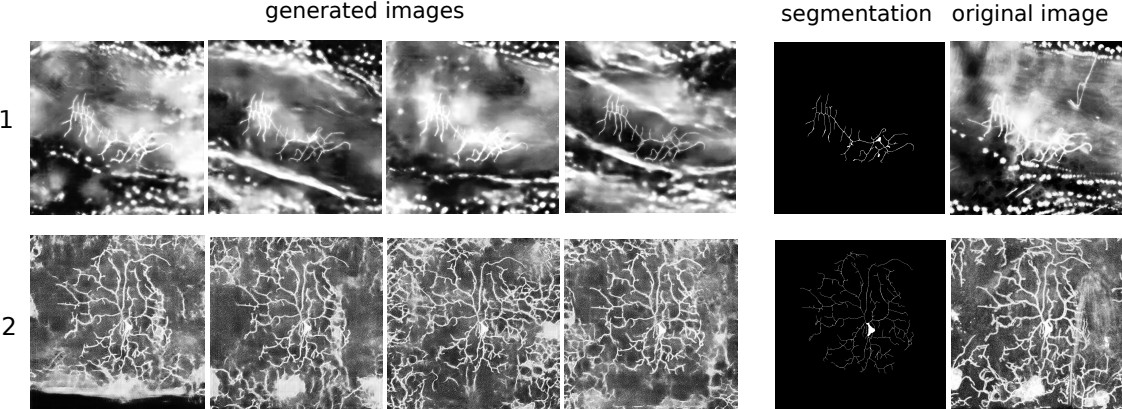

**Figure 6:** Different images generated by our method from a single input segmentation. The row 1 is an example from Dataset 1, the row 2 an example from Dataset 2.

scale 0, but the generated images are refined using the strategy of (Uzunova et al., 2019), (3) the proposed model without the $\mathcal{L}_{MAE}$ loss, and (4) the proposed model. We keep the same hyperparameters and network architecture for all models. The realism of the generated images is measured by the Frechet inception distance at the first scale $FID_0$, and at the final scale $FID_n$. The *scale_corr* metric measures the coherence of the generation process across scales, *repeat_corr* reflects the diversity of the generated images, and *label_corr* shows the compliance with the input neuron morphology. The evaluation metrics are further described in Appendix C. Table 1 shows the results of this ablation study. We show examples of generated images in Figure 8 of Appendix D.

| model | $FID_0 \downarrow$ | $FID_n \downarrow$ | $scale\_corr \uparrow$ | $repeat\_corr \downarrow$ | $label\_corr \uparrow$ |
|---|---|---|---|---|---|
| (1) pix2pix + (Uzunova et al., 2019) | 0.00 | 0.66 | $0.98 \pm 0.01$ | $1.00 \pm 0.00$ | $0.66 \pm 0.05$ |
| (2) cGAN with $\mathcal{L}_{ms}$ + (Uzunova et al., 2019) | 0.29 | 10.04 | $0.04 \pm 0.03$ | $0.45 \pm 0.06$ | $0.78 \pm 0.02$ |
| (3) ours without $\mathcal{L}_{MAE}$ | 0.27 | 0.71 | $0.66 \pm 0.06$ | $0.31 \pm 0.08$ | $0.66 \pm 0.02$ |
| (4) ours | 0.26 | 0.38 | $0.97 \pm 0.01$ | $0.19 \pm 0.08$ | $0.70 \pm 0.03$ |

**Table 1:** Ablation study: our method enhances the diversity of the generated images while keeping a high fidelity to real images.

The multi-scale pix2pix model (1) generates realistic images, but the model is deterministic ($repeat\_corr = 1$). As shown by the *label_corr* values, the generated images do not strictly follow the input neuron as non-annotated branches (e.g. axons) are generated in the background. The high $FID_n$ score of the model (2) supports our assumption that the refinement strategy of (Uzunova et al., 2019) relies on the one-to-one mapping to the real images and is not compatible with the generation of diverse images. The mode seeking loss enhances the diversity of the generated images and improves the compliance to the input neuron morphology. Finally, the $\mathcal{L}_{MAE}$ loss of the proposed model helps to maintain the coherency of the generation across scales, increasing the final image diversity without compromising realism.

### 3.3. Neuron segmentation

In this section, we evaluate the applicability of our generation method for neuron segmentation, which is often a prerequisite to tracing algorithms (Zhou et al., 2018). To this aim,

we prepared four datasets; real images, images generated by our method from the training neurons of Dataset 1, images generated from the unseen neurons of NeuroMorpho 1, images generated by adding noise according to the method of (Radojević and Meijering, 2019). We train a U-Net model to segment the neurons on a given dataset and infer on another dataset (training details in Appendix B). We compute the Dice score of the predicted segmentation to measure the tranferability between generated and real images. The results can be found in Table 2.

| test \ train | real | generated Dataset 1 | generated NeuroMorpho 1 | Dataset 1 + noise |
|---|---|---|---|---|
| real | $0.71 \pm 0.09$ | $0.59 \pm 0.11$ | $0.59 \pm 0.11$ | $0.30 \pm 0.11$ |
| generated Dataset 1 | $0.79 \pm 0.03$ | $0.90 \pm 0.01$ | $0.88 \pm 0.02$ | $0.40 \pm 0.17$ |
| generated NeuroMorpho 1 | $0.76 \pm 0.08$ | $0.87 \pm 0.06$ | $0.89 \pm 0.07$ | $0.42 \pm 0.16$ |
| Dataset 1 + noise | $0.16 \pm 0.15$ | $0.02 \pm 0.03$ | $0.03 \pm 0.02$ | $0.76 \pm 0.09$ |

**Table 2:** Transferability study: neuron segmentation Dice scores obtained by transfer between real and generated datasets.

The images generated by our method show a high transferability to real images compared to the noise-based synthetic images. The gap in Dice score compared to the training on real images is caused by some false positives in the background, missing cell bodies, and segmentation of the axons absent in the ground-truth (Figure 9 of Appendix D for visualizations). We can expect that tracing models trained on images generated using the proposed method generalize better to real data than the current models trained exclusively on noise-based synthetic data (Chen et al., 2021; Prabhakar et al., 2024). Furthermore, our model maintains a good generation quality and diversity even when trained on small datasets (Table 5 of Appendix D), limiting the need for annotated data. Besides, as demonstrated in this study, and supported by Figure 7 and Table 4 of Appendix A, it is able to generalize to input neuron morphologies that differ from the the training set. Therefore, it can find a potential application in cases where manual tracing can hardly be performed (e.g. intricated neurons with complex branching patterns). Indeed, our model only requires segmentations to generate images, which are less burdensome to produce than tracings. The paired ground-truth tracing and microscopy images could then be created by using other available tracing data.

## 4. Conclusion and discussion

We have developed a novel method utilizing a multi-scale cascade of conditional GANs to generate a diverse set of realistic, high-resolution microscopy images from neuron tractograms. Given its ability to provide accurate and rich ground-truth information, including the underlying topology and orientation of the neuronal trees, our method holds promise for facilitating more complex analyses such as tracing. Looking forward, we aim to extend our methodology to other tree-like structures, such as vessels and airways, and to explore its applicability across different imaging modalities. A limitation of our current approach is the reliance on histogram equalization of images, a pre-processing step that, while beneficial for the data used in this study, may not be suitable for all applications. In future work, we intend to explore the generation of images with intensity distributions that more closely mirror those found in real datasets. Additionally, we aim to enhance the interpretability of our generative model, striving for finer control over the generated images' characteristics.

## Acknowledgments

This work was funded by the Japan Society for the Promotion of Science (JSPS, Fellowship P23757). It is supported by the program for Brain Mapping by Integrated Neurotechnologies for Disease Studies (Brain/MINDS) and the program for Multidisciplinary Frontier Brain and Neuroscience Discoveries (Brain/MINDS 2.0) from the Japan Agency for Medical Research and Development AMED (JP23wm0625001, JP15dm0207001).

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

## Appendix A. Dataset details

The images from Dataset 2 were collected from animals at various stages throughout larval development. About half of the images are of uninjured neurons, and half of the images are of neurons recovering from injury to their dendrites. Injury had been delivered by a 2-photon laser either 24 hours prior to imaging, or 72 hours prior to imaging. It has been previously demonstrated that injury causes alterations to dendrite morphology (Thompson-Peer et al., 2016).

## Appendix B. Training details for the segmentation model

Each dataset is divided into train/validation/test folds containing respectively 70%, 15%, and 15% of the data. The number of images in each fold for the different datasets is 291/64/62 for the real images, the noise-based synthetic images, and the generated Dataset 1 images, and 259/57/57 for the generated NeuroMorpho 1 images. The architecture of our segmentation model is a U-Net. We train on patches of size $256 \times 256$, with a 50% overlap. The training loss is a combination of BCE loss and Dice loss. We use the Adam optimizer with a learning rate of 0.01 and a batch size of 64. The training is automatically stopped based on the validation loss, and the best model is retained. At inference, the overlapping patches are merged with cosine weights, and the probability map is thresholded at 0.5 to compute the Dice score.

## Appendix C. Evaluation metrics

The details of the implementation of the metrics used in Section 3.2 are given hereafter.

- $FID_0$ : Frechet inception distance (FID) between the low resolution (64x64) images generated by the base generation cGAN at scale 0, and the real images, downsampled to the same resolution (64x64).

- $FID_n$ : Frechet inception distance (FID) between the full-resolution images generated at the last scale $n$, and the real images. $FID_0$ and $FID_n$ were computed using the python package torchmetrics.

- $scale\_corr$ : Mean cross-correlation between the images generated at different scales. The mean cross correlation value is averaged over 10 different input segmentations.

- *repeat_corr* : Mean cross-correlation between images generated for 10 repetition of the same input. The mean cross-correlation is averaged over 10 different input segmentations.

- *label_corr* : Mean correlation between the input segmentations and the generated images. To be independent of the background intensity, the correlation is computed on local patches (10x10) along the neurites, and averaged.

## Appendix D. Additional figures

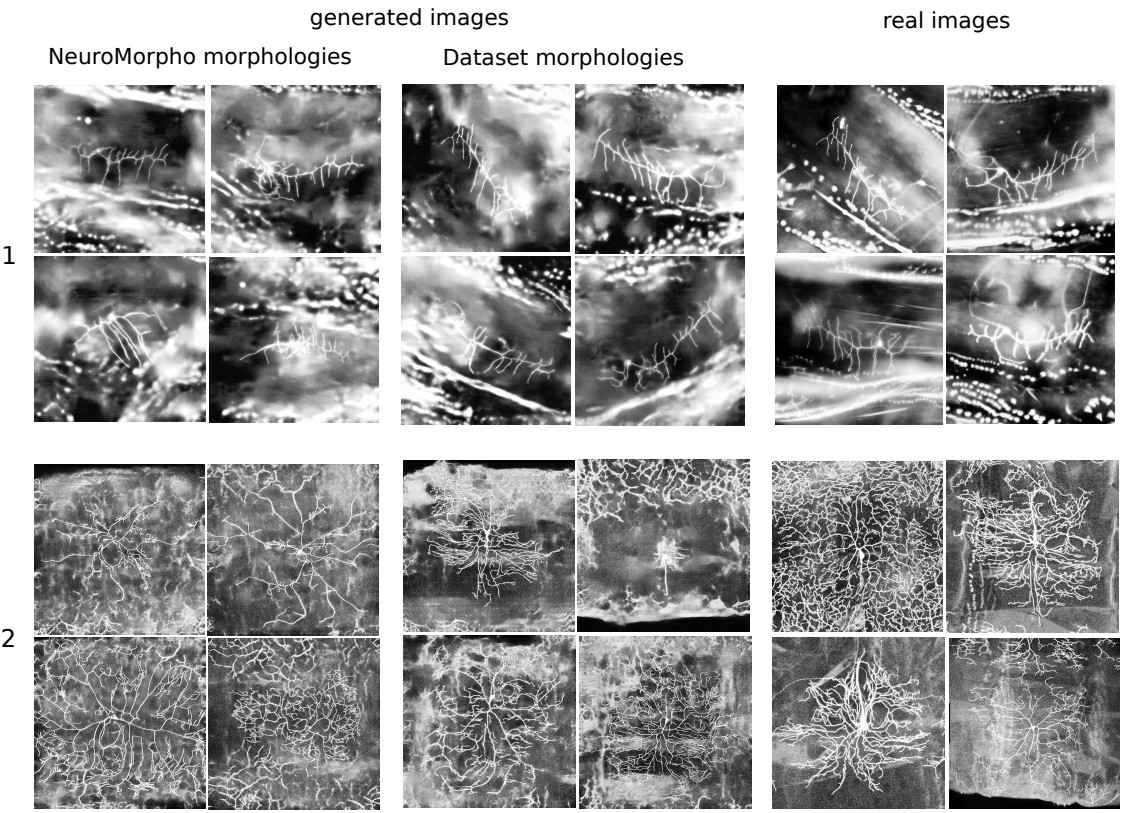

**Figure 7:** Examples from different datasets; images generated by our method with neuron morphologies from the NeuroMorpho dataset (NeuroMorpho 1 and 2), images generated by our method with neuron morphologies from the train dataset (Datasets 1 and 2), and real images (after histogram equalization).

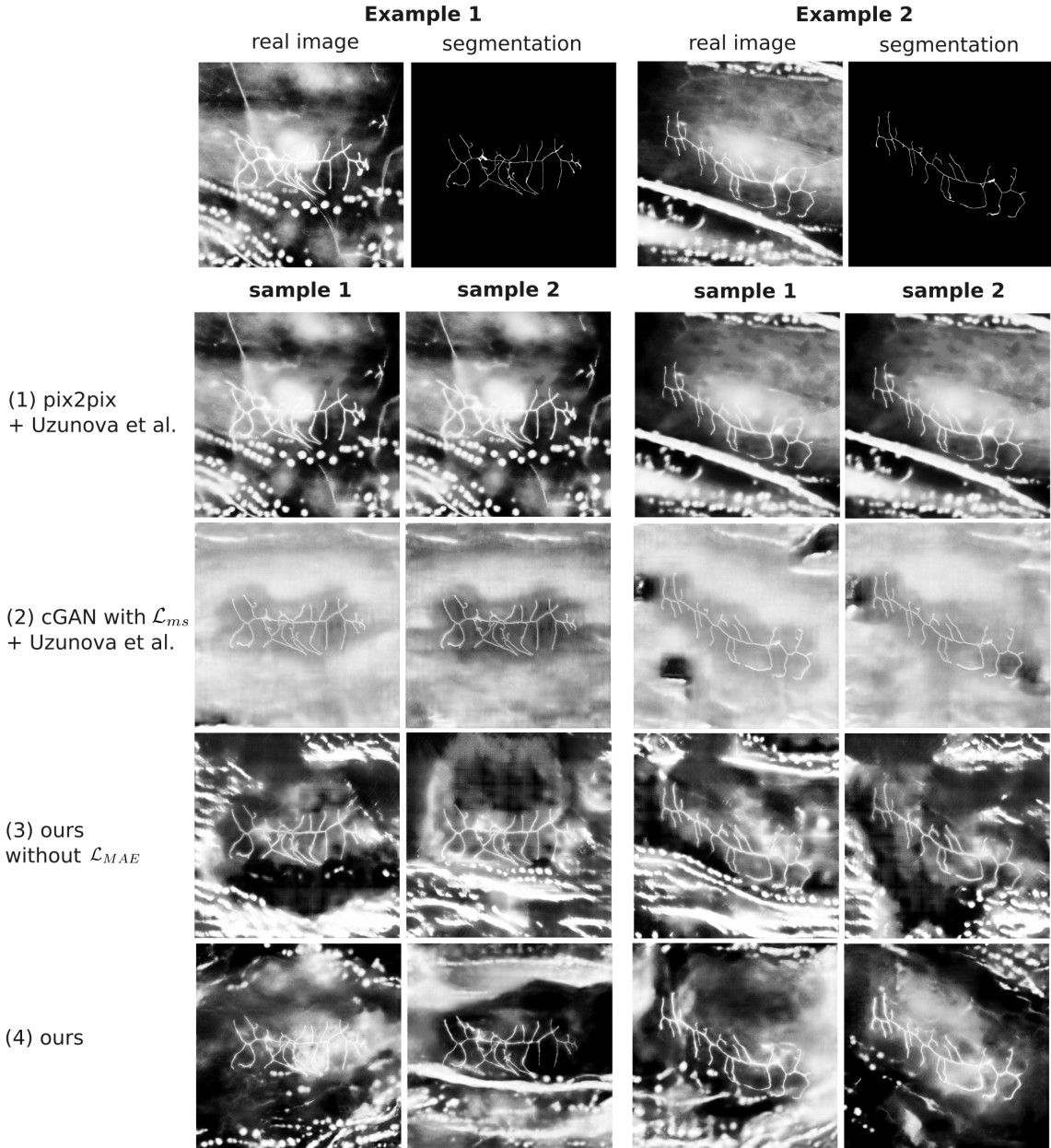

**Figure 8:** Images generated by the models tested in our ablation study. For each model, two samples generated from the same neuron segmentation input are displayed in order to visualize the diversity of the synthetic images.

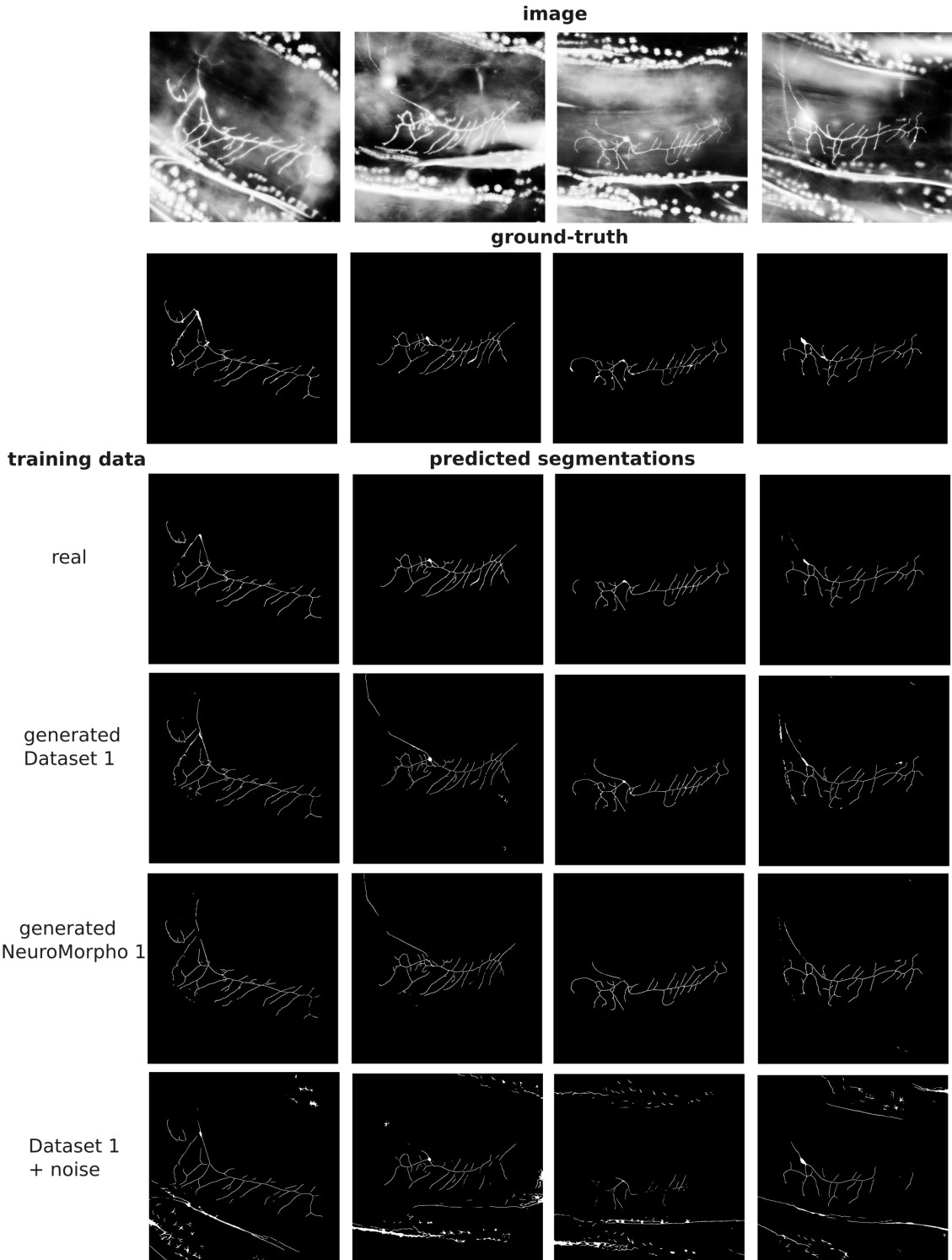

**Figure 9:** Neuron segmentation predicted from real microscopy images by models trained on different synthetic datasets.

| scale | 0 | 1 | 2 | 3 | 4 |
|---|---|---|---|---|---|
| receptive field size / patch size | 1024 / 64 | 512 / 64 | 256 / 64 | 256 / 128 | 256 / 256 |
| resolution factor | 1/16 | 1/8 | 1/4 | 1/2 | 1 |
| iterations | 3000 | 3000 | 6000 | 6000 | 6000 |

**Table 3:** Parameters used for training at each scale of the generation process. The receptive field size (i.e. the amount of the original image contained in the patch) and the patch size are expressed in number of pixels. The resolution factor refers to the ratio between the resolution of the original image at the resolution of the image generated at a given scale. A resolution factor of 1 indicates an image at full resolution. The number of epochs is expressed relatively to the total number of iterations (dataset size × batch size), to make it independent from the size of the input dataset.

| training | inference | $FID_n \downarrow$ | $repeat\_corr \downarrow$ | $label\_corr \uparrow$ |
|---|---|---|---|---|
| Dataset 1 | Dataset 1 | 0.38 | $0.19 \pm 0.08$ | $0.70 \pm 0.03$ |
| Dataset 1 | NeuroMorpho 1 | 1.10 | $0.16 \pm 0.04$ | $0.75 \pm 0.02$ |
| Dataset 2 | Dataset 2 | 1.20 | $0.37 \pm 0.13$ | $0.76 \pm 0.05$ |
| Dataset 2 | NeuroMorpho 2 | 1.08 | $0.43 \pm 0.16$ | $0.82 \pm 0.04$ |

**Table 4:** Performance of the generation model for inference on NeuroMorpho neuron morphologies unseen during training. The images generated from NeuroMorpho neurons show similar levels of diversity, compliance with the input neuron morphology, and fidelity to the real images as the images generated from the training neuron morphologies. The bigger gap in $FID$ observed in Dataset 1 can be explained by the important differences between the neuron morphologies of Dataset 1 and NeuroMorpho.

| training set size | 10 | 20 | 50 | 100 | 291 |
|---|---|---|---|---|---|
| $FID_n$ | 1.17 | 0.57 | 0.46 | 0.30 | 0.38 |
| $repeat\_corr$ | $0.50 \pm 0.19$ | $0.40 \pm 0.10$ | $0.26 \pm 0.12$ | $0.29 \pm 0.07$ | $0.19 \pm 0.08$ |

**Table 5:** Impact of the size of the training set on the model performance. The decrease in the realism and diversity of the generated images observed when we reduce the number of training pairs remains within an acceptable range even with few data (20-50).

