# OpenReview forum: "Multi-scale Stochastic Generation of Labelled Microscopy Images for Neuron Segmentation"
_MIDL.io/2024/Conference — MIDL 2024 Poster_

### Official Review · Reviewer_oN5a · 2024-02-25

**Confidence:** 5
**Preliminary Rating:** 2
**Final Rating:** 3.5

**Summary:**

The authors present a novel method based on conditional GANs to create synthetic neuromorphologies. They utilize two datasets from Drosophila neuromorphology with ca. 400 and 200 images each. The authors present their method tailored to neuromorphologies, provide examples of their cGAN approach and showcase the training with synthetic and real data. The authors provide evidence, that synthetic data alone creates acceptable segmentations, however, when real data is added, it still outperforms the fully synthetic set. The authors show an interesting, but not surprising approach to increase scarce data in the field of neuroanatomy.

**Strengths:**

I like the clear explanation of their cGAN approach, and the formalized loss functions. This clearly shows the single aspects of the cGAN and provides an in-depth explanation of their work. Further, they relate their work nicely to related works that have dealt with tree-like data, such as retina vessels or airways.

**Weaknesses:**

- Weak related works neuroanatomy studies. Generative AI, also cGANs, have been used in combination of neuroimaging studies that involves MRIs. In addition, there are very recent studies that focus on neurons and dendrite segmentations. I'd recommend mentioning those.
- The new method is nicely described, however, I haven't observed any studies how the single components are necessary, i.e. no ablation study has been performed or hyperparameters systematically and quantitatively tested. The authors claim that "the generator often produced structures absent from the base images", one example of anecdotally mentioned results. I would recommend showing the effects (if necessary, use the appendix)
- The authors very often describe features and methods that they use with a rationale intransparent for the reader. Is for example the seeking loss essential?
- The reverse neuron segmentation pipeline is nice. However, in the summary statistics in Table 2 I am missing standard deviations and more examples. For example, in Figure 6, I would like to see the segmentation quality and how this behaves on real and synthetic data.
- I am not getting why all synthetic data in Figure 6 looks alike. Is there almost no variation to existing, real images?

**Detailed Comments:**

Please check the manuscript on missing brackets, missing spaces and missing a**n** in front of letters, such as *an L1* loss. Further, L1 loss is an MAE loss, please use consistent terms and mathematical equations to make the study easier readable for others. I haven't seen any code provided on Github or similar, and the datasets are not linked to their public repository. If the data is not public available, statements where the data is coming from is important. Further, I haven't seen that any optimizer was mentioned in the study. The learning rates are rather high for Adam, this is why it would be very interesting which optimizers the authors used.

**Justification Of Final Rating:**

The authors replied and answered a lot of my questions and improved the manuscript; I am slightly now in favor of the manuscript being seen at MIDL compared to my prior rating. However, my engagement is still borderline.

**Justification Of The Preliminary Rating:**

I believe the study is of merit and potentially interesting to MIDL. I believe that this study, however, needs more work to not only provide a given method without knowing exactly, why the key components are chosen and if they add relevant performance boosts. The limited results and analysis can be largely improved to gain more benefits.

**Questions To Address In The Rebuttal:**

* Clear explanation and validation of the key method components
* Showing variability and neuron segmentation performance
* Elaborate on training statics and outcomes, incorporate more literature w.r.t. neuroanatomy, segmentation and synthetic data.

**Special Issue:**

No

---

> ### Author Response · Authors · 2024-03-18
> **Answers to reviewer oN5a**
>
> Thank you for your comments. We have revised our article in light of your suggestions.
>
> **Q : Weak Related Work on Neuroanatomy Studies using cGANs**
>
> **A :** We have added several references to previous work employing cGANs to generate images from different modalities (MRI, CT, Ultrasound) in the introduction.
>
> **Q : Lack of Ablation and Quantitative Evaluation.**
>
> **A :** As suggested, we have conducted an ablation study. We show the results in an additional section (Section 3.2), as well as visual examples of images generated by each tested model (Appendix D). This study demonstrates the contribution of the different components of our model, as well as the improvements of our approach compared to other state-of-the-art models. Besides, it enabled us to identify a redundancy between the MAE loss used in the refinement model and the training strategy consisting in refining simultaneously the real images. We have found the MAE loss to be sufficient to maintain the diversity and the realism of the generated images. We have modified the model in consequence (Figure 4, Section 2.2.3).
>
> **Q : Is the Mode Seeking Loss Essential?**
>
> **A :** We quantified the contribution of each component of the method in the ablation study of Section 3.2. In particular, we demonstrated the importance of the mode seeking loss to enhance the diversity of the generated images.
>
> **Q : Segmentation Experiment; Standard Deviations and Visual Examples.**
>
> **A :** We have modified the original segmentation pipeline with a new experiment focusing on the applicability of the generated data to segment real data (Section 3.3). The standard deviations are included in the new Dice score result table (Table 2). We have also added visual examples of the segmentation predicted after training on real and synthetic datasets (Appendix D).
>
> **Q : Diversity of the Real and Generated Images?**
>
> **A :** The diversity of the generated images shown in Figure 6 mirrors the variations observed in the training data. We have quantitatively evaluated the diversity of the generated images (Table 2). Examples in Figure 7 of Appendix D demonstrate the diversity in real and generated images.
>
> **Q : Detailed comments : Naming of the MAE loss, Availability of the Data and Code, Optimizer.**
>
> **A :** We have renamed the L1 loss LMAE loss as suggested, proof-read and simplified the writing of the equations in the article. The code will be available after publication of the article. We have added the corresponding github link to the end of the introduction. The datasets are not publicly available, we have cited the corresponding papers to state the provenance of the data, and provided more details on the provenance of the data (Appendix A). The optimizer used for the training is Adam, this information was added in the training details section.

---

> > ### Comment · Reviewer_oN5a · 2024-03-19
> >
> > Thank you for adding the extra validation, experiments and changes in the manuscript.

---

### Official Review · Reviewer_NiU1 · 2024-02-27

**Confidence:** 4
**Preliminary Rating:** 3
**Recommendation:** Poster
**Final Rating:** 4

**Summary:**

This work presents a method based on conditional GANs for the generation of new microscopy samples from single neuron tractograms, in an effort to address the lack of annotated data for the task of neuron dendrite tracing and segmentation. A multi-scale cascade process is adopted, iteratively increasing the resolution and refining the generated samples. The authors show generation results on two different datasets, and evaluate the effect of the addition of the synthetic samples into the training of a standard semantic segmentation model.

**Strengths:**

- The definition of the conditional GAN approach and different modules and loss functions utilized is well done. The loss functions and/or architecture choices are often explained by previous work and/or preliminary experiments.
- The technical writing is generally good and clear.

**Weaknesses:**

- There is a lack of relevant baselines in the evaluation of the boost in performance when adding the synthetic samples into the segmentation training. The authors compare the enhanced (real + synthetic) setting to the vanilla real setting, but having baselines with more standard data augmentation approaches (real + data aug.) would have been important here.
- The authors refer to diversity of generated samples often in the manuscript, but do not discuss on how they assess or evaluate (qualitatively and ideally quantitatively) diversity. There is currently a lack of robust metrics to assess the quality and diversity of image generation models. Regularization in the loss function(s) that generates more diversity is one way of mitigating it, but most of the diversity claims from the authors seem to be purely qualitative. For example, in Figure 3, there is a lot of visual redundancy in the generated samples.

**Detailed Comments:**

- I am a bit surprised that the authors do not cite/mention any diffusion models in the related work section. There has been a huge body of work applying diffusion models for image generation in the last 2 years, and I would expect some of this work would have tackled generation of microscopy images. For example: https://www.biorxiv.org/content/10.1101/2023.07.06.548004v1.full.pdf.
- It would have been useful to show more examples of generated samples. There is only one generation figure in the main text (Fig. 5), and it's mostly there to present the multi-scale process. The extra figure in the Appendix should have been included in the main text.
- Would this cGAN approach be easy to replicate for other use cases in microscopy? It is unclear how much transferability this can have outside of the dendrite images.

**Justification Of Final Rating:**

I am improving my rating since the authors addressed my two main concerns: (1) the lack of quantitative validation metrics for the sample generation methodology, and (2) an improved segmentation section with comparison to baselines as well as ablation studies.

**Justification Of The Preliminary Rating:**

My main concern is the lack of other baselines in the segmentation section in order to demonstrate that the synthetic samples are more useful than a more standard data augmentation approach. The authors claim that their cGAN method will help addressing the lack of labelled datasets for segmentation, so it would have been important to show that data augmentation alone is not as good as their synthetic generation approach. The first part of the paper (the cGAN) is good.

**Questions To Address In The Rebuttal:**

- The authors do perform experiments in order to demonstrate that the synthetic images slightly improve performance for segmentation tasks (when compared to real images only). However, one big question mark is whether they would have observed the same (or better) increase in performance simply by adding an extensive data augmentation strategy to the real samples during training? They do mention using some data augmentation for the image generation, but I am assuming that no data augmentation is used in the segmentation experiments. They cannot justify using their cGAN approach if they do not demonstrate that it offers better performance compared to the standard data augmentation techniques.
- In the refinement cGAN, why is lambda set to 0.1? Does this come from tuning?
- Similarly, in the training details, some hyperparam. choices are not explained. Are these based on previous work, or determined from hyperparam. tuning?

**Special Issue:**

No

---

> ### Author Response · Authors · 2024-03-18
> **Answers to Reviewer NiU1**
>
> Thank you for your comments. We have revised our article in light of your suggestions.
>
> **Q : Lack of Baselines with Standard Data Augmentation?**
>
> **A:** We explored augmentation, finding it improved segmentation results for both real images and when applied to generated data. For training on 10 images, the Dice scores were: real only 0.619 +- 0.109, real + data augmentation 0.641 +- 0.091, generated + fine-tuning 0.642 +- 0.098, and generated + fine-tuning + data augmentation 0.652 +- 0.09.
> However, since the generated images led only to minor improvements, we decided to replace this experiment with an ablation study (Section 3.2) and another, simplified segmentation experiment (Section 3.3). We believe these new studies better highlight the novelty and advantages of our generation model. The new segmentation experiment, where a segmentation model is trained on synthetic datasets and tested on real data, demonstrates through quantitative (Dice score) and qualitative results (Figure 9, Appendix D) that models trained on generated images generalize well to real images.
>
> **Q: Evaluation of Sample Diversity?**
>
> **A :** The lack of quantitative metrics was indeed a limitation of the article. We addressed this limitation using four quantitative metrics to evaluate the proposed approach, including a metric measuring the diversity of the images created. We included an ablation study (Section 3.2), showing that the diversity of the generated images is greatly improved compared to deterministic models like pix2pix, without compromising the realism of the images.
>
> **Q : What About Diffusion Models?**
>
> **A :** Diffusion models have indeed been successfully applied to image generation in the past years and we believe that they could contribute to improving our model. To the best of our knowledge diffusion models have not yet been applied to the conditioned generation of microscopy images. The work cited in the comment employs diffusion models to generate unconditioned artificial microscopy images of microtubules. This lack of conditioning makes the ground-truth information unavailable, unlike the proposed method. Nevertheless, we have identified some recent work on conditioned diffusion models from the computer vision domain that we would like to apply to microscopy image generation in a future work [2].
>
> [2] *Ho, Jonathan, et al. "Cascaded diffusion models for high fidelity image generation." Journal of Machine Learning Research 23.47 (2022): 1-33.*
>
> **Q : More Examples of Generated Samples?**
>
> **A :** In addition to Figure 5, Figure 6 also shows some examples of generated images (4 generates images for each dataset) in the main text. Unfortunately, due to space limitations, we could not move the figure from the appendix to the main article. However, we added a new figure to Appendix D (Figure 8), presenting the images generated by the different models included in our ablation study.
>
> **Q : Application of the Method on Other Use Case in Microscopy?**
>
> **A :** As shown in the article, our approach was able to generate realistic images from two datasets showing very different neuron morphologies and imaging artifacts. Although we have not tried yet to generate other types of microscopy images, the results look promising. In the present article, we have chosen to condition the image generation on the neuron dendrites for an application to neuron tracing. However, we could imagine different applications, for instance, in vessel segmentation by conditioning the model on the vessel masks [1], or in cell counting by conditioning it on the cell positions.
> Beyond microscopy images, we believe that the proposed method can be transferred to other imaging modalities showing tree-like structures such as MRA (vessel) and lung CT scan (airways). Attempting to generate a broader range of imaging modalities is the challenge that we will tackle next.
>
> [1] *Poon, Charissa, et al. "MiniVess: A dataset of rodent cerebrovasculature from in vivo multiphoton fluorescence microscopy imaging." bioRxiv (2022): 2022-07.*
>
> **Q : How Were the Value Lambda, and Other Hyperparameters Decided?**
>
> **A :** There was a mistake in the article, the original lambda parameter was set to 10. We chose this value because it roughly corresponds to the ratio between the adversarial loss and the MAE loss, giving similar importance to both losses in the refinement process. In our experiments, we have found the optimal lambda value to be between 10 and 50. These values give enough freedom for the generator to create new details and correct the mistakes of previous refinements while keeping a high coherence between scales and therefore preserving the diversity observed at scale 0. The impact of the lambda parameter on the generation is illustrated in the ablation study (Section 3.2, Table 1). The other hyperparameters were tuned by experimenting with different values heuristically. We have mentioned this in the training details section.

---

> > ### Comment · Reviewer_NiU1 · 2024-03-20
> >
> > I thank the authors for the adjustments they made to the manuscript, and particularly the addition of evaluation metrics for the image generation, and an ablation study for the segmentation section.

---

### Official Review · Reviewer_1Ypj · 2024-02-28

**Confidence:** 5
**Preliminary Rating:** 2
**Recommendation:** Poster
**Final Rating:** 4

**Summary:**

Decroocq et al. propose a conditioned generative approach to generate virtual microscopy images from a given binary mask. The method builds upon previous work from Uzunova et al. 2019 (cited in the manuscript) to iteratively reconstruct the generated image using a hierarchical schema. To support the generation of different synthetic images for a given mask, they introduce a regularising random component together with the input mask as suggested by Mao et al. 2019 (cited in the manuscript). The approach is explained in detail. The method is used with two different datasets but quantitative metrics are only shown for Dataset I (NeuroMorpho dataset of Drosophila class I).

**Strengths:**

- Labelled microscopy image data generation is still an underexplored task in biomedical imaging in terms of finding accurate and reproducible approaches. In this sense, the authors propose a method relatively easy to replicate and which seems to provide accurate virtual images.
- The architecture of the proposed approach is described in detail.
- The results obtained seem to be realistic.
- The authors considered the final goal of this approach (increasing training datasets for neuron segmentation) in their assessment and evaluated the impact of synthesising labelled microscopy images when training a network for segmentation.

**Weaknesses:**

- The proposed approach requires a tiling strategy to crop different patches of the generated images at different scales and enhance their resolution. Yet, **there are no details about how this tiling is performed**.
   - Is the receptive field of the generator taken into consideration?
   - How is the overlap between patches performed to avoid tiling artefacts?

- In Figure 1, it seems that in the Scale 1, a patch is cropped from the generated image and then downsampled. This downsampled patch is refined and then upsampled again to repeat the process. Is this correct? Then, why do the authors upsample the refined image, if this one is then downsampled? Also, by downsampling the patch, there is less space in pixels to improve the resolution of the generated image.
   - When training the cGANs, what's the resolution of the ground truth image? Are these downsampled accordingly?

- In the text, the authors use the concept of resolution to refer to scaling (super-pixelating) factors. Please, note that **resolution is not the size of an image in pixels**, which is given in Table 1. Also, the description in section 2.2.4 should be more clear: why does the number of training iterations change for different scales? There seems to be a relationship between the patch size of the input and output images but this is not clear in the text. Does it mean that the hierarchical steps are limited to a few upsampling steps?

- While the proposed approach looks promising, **the current work lacks proper quantitative assessment**:
    - The authors do not provide any quantitative metric to assess the quality of the generated images, nor about the training process. Because this is a conditional GAN, there are ways of computing pixel-wise metrics. Also, there are ways of measuring the quality of the generated images with respect to the ground truth domain using different perceptual metrics.
    - The results in Table 2 cannot be interpreted without standard deviations. Indeed, looking at the three different combinations, one would conclude that the generated images do not contribute to the learning process of the network as the results for the real and generated+real are almost the same.

- **The entire segmentation pipeline should be better described**:
    - In section 2.3, it is said "At inference, we generate 100 synthetic images per training subset from the segmentations". What does this mean? Are these the images generated to train/fine-tune the network? If this is the case, it would explain why the generated images do not contribute to the learning process, as in the end, the masks used are the same as the real ones.
    - What does it mean that training subsets were taken? Is it that the authors take only 10 images from the training dataset and train a UNet only using those 10 images? then, why are not the 291 images taken?
    - Also, what's the hierarchical scaling chosen for this evaluation?

**Detailed Comments:**

All the comments and detailed suggestions are described in the weaknesses section.

**Justification Of Final Rating:**

Annotated training data generation is quite an interesting approach for bioimaging. In this sense, the work of the authors becomes interesting.
The revised version of the manuscript addresses all the points highlighted in the weaknesses of this review. Importantly, the authors' methodology and metrics to test the accuracy of the proposed method (image generation and posterior testing with a segmentation task) are convenient and a good reference for future works.

Still, the improvement of the segmentation accuracy is unclear, so an example in which this method can be of important utility is missing, which one could agree that it is out of the scope of the current paper/conference.

**Justification Of The Preliminary Rating:**

While the method proposed by the authors looks promising, it lacks proper quantitative assessment and the manuscript needs to improve some of the descriptions. I would recommend the work to be accepted as long as these are addressed.

**Questions To Address In The Rebuttal:**

The questions to address in the rebuttal are already specified in the previous Weaknesses section.

**Special Issue:**

No

---

> ### Author Response · Authors · 2024-03-18
> **Answers to Reviewer 1Ypj**
>
> *1. Tiling strategy*
>
> **Q: Receptive Field Consideration?**
>
> **A:** The receptive field of the generator was not taken into consideration when deciding the patch size. It was set so that the resolution of the image is doubled at each scale. We have found this strategy to be a good trade-off between the difficulty of the refinement model’s task (i.e. the resolution gap between the input and output) and the number of models to train. Conducting a study of the impact of the number of scales and patch sizes would be an interesting direction for further experiments.
>
> **Q: Overlap Handling to Avoid Artifacts?**
>
> **A:**  We have detailed the patch strategy employed to avoid border effect in section 2.2.4. We use a 50% overlap between patches, and the overlapping parts are merged based on cosine weights.
>
> **Q: Purpose of Upsampling Before Downsampling?**
>
> **A:** To maintain a low memory footprint, we keep the array size of inputs and outputs for the generator constant at 64^2 while increasing the pixel resolution from coarse to fine. This process effectively decreases the receptive field size relative to the entire image. Initially, a 64^2 patch represents a downscaled version of the entire 1024^2 image. As the resolution increases, the receptive field narrows with each iteration, eventually matching the 64^2 patch size in full resolution. We have modified Section 2.2.1 to clarify this point.
> Figure 1 was misleading concerning the patch upsampling and downsampling in the refinement scales. The 64^2 patches generated at scale 1 are first merged to form the generated 128^2 image. It is this generated image that is upsampled to full resolution, cropped into patches with a smaller field of view, and then each patch is downsampled to 64^2. We have modified Figure 1 to clarify this point.
>
> **Q: Ground Truth Image Resolution and Downsampling?**
>
> **A:** When training the cGANs, ground-truth images are patched and scaled to match the target resolution of the generated images, as emphasized in the modified Figure 1.
>
> **Q: Clarification on Resolution vs. Size**
>
> **A:** The word resolution was misused in Table 1. Besides, both Figure 1 and Table 1 were misleading the reader into thinking that the patches were refined by using a larger output size compared to the input size in the generator. In our method, the refinement is not based on the difference in size between the input and output patches but on the difference in resolution between the input and output patches. We have clarified this point by modifying Table 1, Figure 1, and Section 2.2.1.
>
> **Q: Training Iterations by Scale?**
>
> **A:** The number of training iterations was decided based on the visual observation of the generated images at different epochs. We employ a different number of iterations at scale 0 as we use different loss functions for the base generation model, and at scale 1, as it bridges the base generation scale 0 and the next refinement scales.
>
> *2. Quantitative assessment*
>
> **Q: Quantitative Metrics for Image Quality and Training?**
>
> **A:** The lack of quantitative assessment was indeed a limitation of our work. To address it, we proposed four metrics to measure the performances of our generation model, described in an additional section 3.2. This evaluation is supplemented by an ablation study that emphasizes the contribution of each component of our model.
>
> **Q: Need for Standard Deviations in Table 2?**
>
> **A:** We added the standard deviation values for the Dice scores in Section 3.3. The segmentation experiment was however modified, as explained hereafter.
>
> *3. Segmentation pipeline*
>
> **Q: Purpose of Generating 100 Synthetic Images?**
>
> **A :** In the original segmentation experiment, we generated 100 synthetic images from real masks to train a segmentation model. The goal was to demonstrate that our stochastic model could enhance segmentation by introducing unseen variations of the original 'real' neuron images into the training set. Although we achieved a slight improvement in segmentation results, the enhancement was not substantial enough to fully demonstrate our model's potential. Consequently, we decided to replace this experiment with an ablation study and a simpler segmentation experiment to better highlight the strengths of our model.
>
> **Q: Reason for Selecting Training Subsets; Hierarchical Scaling?**
>
> **A:** In the original segmentation experiment, our objective was to show that our method can improve the segmentation results in cases where few annotated data is available. To this aim, we trained the model on small subsets of the training set (10, 20, 100). However, in the revised version, we have decided to change this segmentation experiment to focus on the transferability of the generated images to segment real images. In this new experiment, we use the whole training fold (291) during training, as suggested in your comment. The hierarchical scaling employed is as written in the training detail section.

---

> > ### Comment · Reviewer_1Ypj · 2024-03-18
> >
> > I thank the authors' efforts in addressing each of the comments and considerably improving the presentation and validation of their work. As shown in the new figures and proven by the quality metrics, the proposed method can accurately generate virtual images resembling the real ones.

---

### Author Response · Authors · 2024-03-18
**General answer to the reviewers**

We thank the reviewers for their insightful comments. Their suggestions helped us to improve the original article, especially concerning the quantitative evaluation of the proposed approach. Please find hereafter a summary of the changes that were made in the revised version. The modified parts are highlighted in red in the revised article.

**Summary of the revisions:**

- We clarified the explanation of the key points of our patch-based strategy (Section 2.2.1, Figure 1).
- We designed and computed metrics to quantify the realism of the generated images, the compliance with the input neuron shape, the coherence of the generation process across scales, and the diversity of the generated images (Section 3.2)
- We conducted an ablation study clearly demonstrating the contribution of the different components of the proposed model (Section 3.2, Table 1). The ablation study enabled us to identify and remove redundant components in our model (Section 2.2.3, Figure 4).
- We replaced the original segmentation section by a new, simpler segmentation experiment focusing on the transferability of the generated images to segment real dataset (Section 3.3, Table 2). The new experiment, together with the ablation study, emphasizes better the advantages of the proposed approach.
- We provided more visualizations of the generated images and the segmentation predictions, as well as additional quantitative results, in appendix (Appendix D).
- We modified the title of the article to underline the originality of the proposed method.

We hope that your concerns were addressed in the revised version of the article and that you will consider us for MIDL 2024 conference.

---

### Author Response · Authors · 2024-03-25
**Thanks to the reviewers**

We would like to thank the reviewers once again, whose comments and suggestions have enabled us to improve the original article. We have uploaded a clean version of the article, without highlights.

---

### Meta-Review · Area_Chair_VsuD · 2024-04-01

**Recommendation:** Accept (Poster)
**Confidence:** 5

**Metareview:**

This paper presents an interesting method for multi-scale image generation. The methods are clearly presented and well motivated. The authors perform extensive experiments and insightful ablation studies.

However, the paper remains hard to follow/unclear about the presentation of the experimental set-up: implementation details, training procedure, lack of a validation set for hyper-parameter tuning, data splits, etc. Moreover, the few given implementation/experimental details are described in a very long appendix, which further worsens readability.

Overall, I think this paper will lead to interesting discussions at MIDL, but I strongly suggest the authors to clarify their experimental set-up.

---

### Decision · Program_Chairs · 2024-04-06

Accept (Poster)